# The impact of electronic health record discontinuity on prediction modeling

Shreyas Kar[1☯], Lily G. Bessette[1☯], Richard Wyss[1], Aaron S. Kesselheim[1], Kueiyu Joshua Lin[1,2]*

1 Department of Medicine, Division of Pharmacoepidemiology and Pharmacoeconomics, Brigham and Women's, Boston, MA, United States of America, 2 Department of Medicine, Massachusetts General Hospital, Harvard Medical School, Boston, MA, United States of America

☯ These authors contributed equally to this work.
* jklin@bwh.harvard.edu

## Abstract

### Background

To determine the impact of electronic health record (EHR)-discontinuity on the performance of prediction models.

### Methods

The study population consisted of patients with a history of cardiovascular (CV) comorbidities identified using US Medicare claims data from 2007 to 2017, linked to EHR from two networks (used as model training and validation set, respectively). We built models predicting one-year risk of mortality, major CV events, and major bleeding events, stratified by high vs. low algorithm-predicted EHR-continuity. The best-performing models for each outcome were chosen among 5 commonly used machine-learning models. We compared model performance by Area under the ROC curve (AUROC) and Area under the precision-recall curve (AUPRC).

### Results

Based on 180,950 in the training and 103,061 in the validation set, we found EHR captured only 21.0–28.1% of all the non-fatal outcomes in the low EHR-continuity cohort but 55.4–66.1% of that in the high EHR-continuity cohort. In the validation set, the best-performing model developed among high EHR-continuity patients had consistently higher AUROC than that based on low-continuity patients: AUROC was 0.849 vs. 0.743 when predicting mortality; AUROC was 0.802 vs. 0.659 predicting the CV events; AUROC was 0.635 vs. 0.567 predicting major bleeding. We observed a similar pattern when using AUPRC as the outcome metric.

### Conclusions

Among patients with CV comorbidities, when predicting mortality, major CV events, and bleeding outcomes, the prediction models developed in datasets with low EHR-continuity

**Data Availability Statement:** The data that support the findings of this study included potentially identifying or sensitive patient information at our institution and cannot be shared outside of our institution based on our Mass General Brigham IRB

(Protocol#2017P002659). Data are available from the Mass General Brigham Institutional Review Boards (contact partnersirb@partners.org) for researchers who meet the criteria for access to confidential data.

**Funding:** Dr. Joshua Lin and this project were National Institute of Health (Grant# R01LM012594 and NIH R01LM013204). Dr. Kesselheim's research is supported by Arnold Ventures. The funders had no role in study design, data collection and analysis, decision to publish, or preparation of the manuscript.

**Competing interests:** The authors have declared that no competing interests exist.

consistently had worse performance compared to models developed with high EHR-continuity.

## Introduction

Evidence based on real-world data is informative for clinical decision-making because the findings are drawn from patients in routine care, including the frail and complex individuals that are often under-represented in clinical trials. Electronic Health Record (EHR) data contain rich clinical data not typically available in other routinely collected data (e.g., insurance claims data) and have been increasingly used for clinical research [1]. However, one drawback to using EHR data is that most US EHR systems do not comprehensively capture clinical encounters from all health care facilities. Thus, an EHR database used in a particular study may miss clinically important records for some patients. We defined EHR-discontinuity as having medical information recorded outside of the study EHR, hence not observable to the investigators. The missing data can lead to misclassification of study variables (i.e., information bias) as it is often difficult to distinguish patient not having the medical condition from the study EHR missing the medical encounter and its associated diagnoses. Prior studies have shown that a single EHR system captures less than 30% of all the healthcare information, which can translate into up to 17-fold greater misclassification of comparative effectiveness research (CER)-relevant variables ascertained based only on EHR in those with an EHR-continuity (i.e., record capture rate) <10% vs. ≥80% [2].

An algorithm was previously developed to identify patients with high EHR-continuity in which the information bias due to EHR-discontinuity can be substantially reduced [3]. However, there have not been any data in the literature assessing how such information bias due to EHR-discontinuity may affect the validity of EHR-based prediction modeling. This is a key knowledge gap since numerous prediction models have been developed using EHR as the primary source [4]. Therefore, we aimed to investigate the impact of EHR-discontinuity on the performance of the prediction models built using EHR data. We applied machine learning (ML) algorithms to predict outcomes of clinical and public health importance, including mortality, myocardial infarction, stroke, and major bleeding events. We compared the model performance when ascertaining predictors and outcomes based on EHR data alone versus models that use EHR supplemented by insurance claims data (EHR+Claims). For patients with medical insurance coverage, we assumed that claims data capture medical information missing from the EHR network. We compared the model performance in the validation set, stratified by levels of EHR-continuity based on a published algorithm [3]. We hypothesized that the model performance developed in patients with high EHR-continuity would be superior to that based on those with low EHR-continuity in the validation set due to less information bias.

## Methods

### Dataset

This study utilized data from the Research Patient Data Repository [5], which contains EHR data from two large multi-center delivery networks in Massachusetts. The EHR data were linked to Medicare fee-for-service Parts A (inpatient coverage), B (outpatient coverage), and D (prescription benefits) claims data by insurance policy number, date of birth, and gender (linkage success rate = 98.7%). The first network—which consists of one tertiary hospital, two community hospitals and 17 primary care centers—was used for model development (i.e., the

training cohort). A second network consisting of one tertiary hospital, one hospital and 16 primary care facilities was used as the validation cohort. Both EHR databases contained information on patient demographics, medical diagnoses, procedures, and medications. The Medicare claims data contain information on demographics, enrollment start- and end-dates, dispensed medications and performed procedures, and medical diagnoses. Since we require the study cohort to have Medicare as the primary payor, we assumed all the covered care is recorded in the Medicare claims database that benchmarks the EHR-continuity in this study. This study was approved by the Institutional Review Board of the Brigham and Women's Hospital, Boston, Massachusetts.

## Study population

The study population was made up of patients aged $\geq 65$ with at least 365 days of continuous enrollment in fee-for-service Medicare Part A (inpatient coverage), B (outpatient coverage), and D (prescription plan) programs and at least one EHR encounter overlapping with the Medicare enrollment period from January 1, 2007, to December 31, 2017. The cohort entry day (CED) was the day when the criteria were met. We further required the study cohort to have at least one of the following cardiovascular comorbidities in the 365 days before cohort entry day: diabetes, coronary artery disease (CAD), systemic embolism, deep vein thrombosis (DVT), pulmonary embolism (PE), atrial fibrillation, hypertension, heart failure, hemorrhagic stroke, ischemic stroke, other stroke effects, transient ischemic attack (TIA), myocardial infection (MI), and peripheral arterial disease (PAD), or dispensation of a related cardiovascular medications (See *S1 Table in S1 File* for definitions of these variables).

## Patient characteristics assessed

We assessed patient characteristics as the candidate predictors using the EHR and claims data in the 365 days before the CED. We assessed the following covariates: 1) demographic variables recorded in the EHR: age, sex, and race. Age was modeled as a continuous variable and race as a multicategory variable with missing indicator for those with unknow race 2) comorbidities based on the diagnosis code (i.e., international classification of disease [ICD] code) and procedure code (i.e., Current Procedural Terminology [CPT] or Healthcare Common Procedure Coding System [HCPCS] codes) and they are: coronary artery disease, venous thromboembolism, hypertension, diabetes, hyperlipidemia, atherosclerosis, heart failure, stroke, myocardial infarction, gastrointestinal and other bleeds, peripheral vascular disease, liver/kidney diseases, dementia. We model these factors as binary variables; 3) prior medication use based on prescribing (EHR) and dispensing (claims) data: aspirin, other antiplatelet agents, nonsteroidal anti-inflammatory drugs, anticoagulants, antihypertensive agents, antiarrhythmics, statins, antidiabetics, acid suppressants; 4) healthcare use variables based on the healthcare encounter information in the EHR and claims data: number of medications, hospitalizations, hospital days, and office visits. These variables are modeled as multicategory variables (see *S1 Table in S1 File* for definitions of these variables).

## Algorithm-predicted EHR-continuity

We used a previously validated algorithm to determine EHR-continuity in the 365 days prior to CED (see model details and performance in *S2 Table in S1 File*) [3]. The model predictors of the EHR-continuity are mainly indicators related to primary care follow-up in the study EHR, including a) codes for a routine-care office visit; (b) preventive interventions or screening tests; (c) recording of diagnoses or medications in the EHR; (d) presence and numbers of certain types of encounters in the EHR; and (e) seeing the same provider repeatedly in the

system (see **S3 Table in S1 File** for variable definitions). Patients classified in the top 20% of predicted EHR-continuity were assigned to the high EHR-continuity cohort, and those in the lower 80% of predicted EHR-continuity to the low EHR-continuity cohort. The cut-off of 20% was informed by a prior study showing the level of misclassification was acceptable in patients with the top 20% of predicted EHR-continuity [3].

### Outcome ascertainment

We built models predicting one-year risk of the following outcomes assessed in the 365 days following (including) the CED: 1) major cardiovascular events (a composite outcome of myocardial infarction, ischemic or hemorrhagic stroke), 2) major bleeding, 3) mortality. The ascertainment of the cardiovascular and major bleeding events was based on the diagnosis and procedure codes in the EHR or claims data (see **S4 Table in S1 File** for variable definitions). The mortality assessment in EHR and claims data was based on discharge status, social security administration information received in the hospital and the death information recorded in the Medicare claims data and national death index. The outcome ascertainment started on the CED until the earliest of the following censoring events: 1) Medicare disenrollment, 2) death, 3) 365 days following CED; 4) end of the study period (2017/12/31).

### Prediction model development and evaluation

The best-performing model for each outcome was chosen among 5 ML modeling methods, including logistic regression (LR), least absolute shrinkage and selection operator (LASSO) regression [6], random forest (RF) [7], gradient boosting (GB) [8], and Bayesian additive regression tree (BART) [9]. The hyperparameters for these prediction algorithms were optimized through grid-search with cross-validation. For each of these modeling methods, we compared the performance with vs. without the following processing techniques: 1) factor analysis of mixed data (FAMD), a dimensionality-reduction technique appropriate for mixed data (i.e., a combination of categorical and continuous variables) [10]; 2) synthetic minority oversampling technique (SMOTE), an oversampling technique to reduce the negative influence of an unbalanced dataset [11] and no preprocessing. We compared model performance by 5-fold cross-validated area under the Receiver Operating Characteristic Curve (AUROC, see details on SMOTE implementation in the **Supplemental materials**). We estimated the optimism (i.e., reduction in performance in external validation datasets when the algorithm is applying to the new subjects) by subtracting the AUROC in the validation from that in the training cohort (a difference of zero or lower would be reported as no optimism). We divided the training and validation cohorts into high vs. low EHR continuity cohorts, respectively. We compared the difference in AUROC in training vs. validation set by EHR-continuity. By level of EHR-continuity, we ascertained two sets of outcomes: the first set based on EHR alone and the second set based on EHR plus claims data. In the validation set, we compare the difference between AUROC when prediction the outcome based on EHR alone vs. that predicting the outcome based on EHR plus claims data. In a sensitivity analysis, we used area under the precision-recall curve (AUPRC) instead of AUROC as the primary model performance metric and repeated all the steps mentioned above. Python (3.9.5) was used for model development and implementation [12].

## Results

### Patient characteristics

Our study population included 180,950 patients (mean age 74.74 years and female 57.1%) in the training cohort and 103,061 (mean age 73.85 years and female 59.4%) in the validation

**Table 1. Patient characteristics of the study population based on EHR and claims data.**

| Characteristics | Training cohort N = 180,950 | Validation cohort N = 103,061 | Standardized difference |
|---|---|---|---|
| Age, years* | 74.74 (7.84) | 73.85 (7.30) | 0.12 |
| Female | 77703 (42.9) | 41834 (40.6) | 0.05 |
| Race | | | |
| White | 151387 (83.7) | 84377 (81.9) | 0.05 |
| Black | 4555 (2.5) | 4117 (4.0) | -0.08 |
| Asian | 3550 (2.0) | 1341 (1.3) | 0.05 |
| North American Native | 152 (0.1) | 98 (0.1) | 0.00 |
| Unknown | 21306 (11.8) | 13128 (12.7) | -0.03 |
| **Medical history** | | | |
| Coronary Artery Disease | 53545 (29.6) | 32354 (31.4) | -0.04 |
| Atrial Fibrillation | 32673 (18.1) | 19496 (18.9) | -0.02 |
| Hypertension | 161729 (89.4) | 92785 (90.0) | -0.02 |
| Heart Failure | 18094 (10.0) | 11705 (11.4) | -0.05 |
| Ischemic Stroke | 13620 (7.5) | 7518 (7.3) | 0.01 |
| Acute Myocardial Infarction | 6019 (3.3) | 4773 (4.6) | -0.07 |
| COPD | 26520 (14.7) | 16561 (16.1) | -0.04 |
| Pneumonia | 16562 (9.2) | 10275 (10.0) | -0.03 |
| Dementia | 20260 (11.2) | 9880 (9.6) | 0.05 |
| Depression | 36300 (20.1) | 20293 (19.7) | 0.01 |
| Peptic ulcer disease | 52787 (29.2) | 31844 (30.9) | -0.04 |
| Cancer | 78710 (43.5) | 52366 (50.8) | -0.15 |
| Falls | 16930 (9.4) | 9204 (8.9) | 0.02 |
| Rheumatoid Arthritis | 5035 (2.8) | 4050 (3.9) | -0.06 |
| Coagulation disorder | 12866 (7.1) | 8595 (8.3) | -0.05 |

EHR = electronic health record, COPD = chronic obstructive pulmonary disease.

cohort. The validation cohort has higher proportion of black race. The prevalence of acute myocardial infarction was higher in the validation than training cohort while the prevalence of most other comorbidities appears to be comparable in the two cohorts (*Table 1*).

## Best-forming model and proportion of outcome events captured by the EHR

The best-performing models varied by the research scenarios without a predictable pattern (*Table 2*). Compared with all the non-fatal outcome events ascertained by EHR and claims data, using EHR only captured only 21.0–28.1% in the low EHR-continuity cohort but 55.4–66.1% in the high EHR-continuity cohort. The difference in under-ascertainment between low vs. high EHR continuity cohort was smaller for the death outcome. The corresponding capture proportions for the death outcomes were 79.0–80.5% in the low EHR-continuity cohort, compared to 84.5–88.5% in the high EHR-continuity cohort (*Table 2*).

## Optimism in models based on high vs. low EHR-continuity

We observed a substantial amount of optimism when the models were developed in those with low EHR-continuity. For example, the AUROC dropped from 0.740 to 0.659 (optimism = 0.081) when predicting one-year major cardiovascular risk and from 0.672 to 0.567

**Table 2. Best-forming model and proportion of outcome events captured by the HER.**

| Continuity | Outcomes | Best-performing model | Training set | | | Validation set | | |
|---|---|---|---|---|---|---|---|---|
| | | | #Event based on EHR alone, N (%) | #Event based on EHR +claims data, N (%) | %Captured by EHR | #Event based on EHR alone, N (%) | #Event based on EHR +claims data, N (%) | %Captured by EHR |
| Low[a] | CV[b] | GBT NP | 8801 (6.1) | 33751 (23.5) | 26.1% | 6152 (7.5) | 21896 (26.8) | 28.1% |
| | Major bleeding | LASSO SM | 4906 (3.4) | 23346 (16.2) | 21.0% | 3677 (4.5) | 15275 (18.7) | 24.1% |
| | Death | GBT NP | 7771 (5.4) | 9832 (6.8) | 79.0% | 5981 (7.3) | 7434 (9.1) | 80.5% |
| High | CV[d] | RF NP | 4969 (13.4) | 7514 (20.3) | 66.1% | 2543 (12.0) | 4246 (20.0) | 59.9% |
| | Major bleeding | GBT NP | 3064 (8.3) | 5116 (13.8) | 59.9% | 1687 (8.0) | 3047 (14.4) | 55.4% |
| | Death | GBT NP | 1479 (4.0) | 1751 (4.7) | 84.5% | 847 (4.0) | 957 (4.5) | 88.5% |

[a] Low vs. high EHR-continuity are those with top 20% vs. lower 80% predicted EHR-continuity based on a published algorithm[3].

[b] Composite cardiovascular events (myocardial infarction or stroke).

EHR = electronic health record, NP = no preprocessing of data, LR = logistic regression, GB = gradient boosting, RF = random forest.

when predicting one-year major bleeding risk in the low-continuity cohort. For a given outcome, the optimism was appreciably smaller in the high EHR-continuity cohort than that in the high EHR-continuity cohort. For example, when predicting one-year major cardiovascular risk, the optimism was 0.081 in the low EHR-continuity cohort but 0.029 in the high EHR-continuity cohort (*Table 3*).

## Model performance in the validation set by EHR-continuity levels

The models developed in the high EHR-continuity persistently have superior performance than that based on the low EHR-continuity in terms of AUROC evaluated in the validation set when predicting outcomes determined by EHR plus claims data (*Table 3*). For example, when predicting one-year major cardiovascular risk, the AUROC was 0.659 in the low EHR-continuity cohort, compared to 0.802 in the high EHR-continuity cohort (Fig 1). In the validation set, the discrepancy between prediction performance based on EHR only vs. EHR-claim data was greater in the low vs. high EHR-continuity cohort. For example, such discrepancy AUROC was 0.091 vs. 0.033 in low vs. high EHR-continuity cohort.

**Table 3. Area under the ROC curve and optimism by EHR-continuity.**

| EHR-continuity | Outcomes | AUROC in the training set | AUROC in the validation set based on EHR-claims data | Optimism[a] | AUROC in the validation set based on EHR only | Discrepancy from the reference standard[b] |
|---|---|---|---|---|---|---|
| Low[c] | CV[d] | 0.740 | 0.659 | 0.081 | 0.750 | 0.091 |
| | Major bleeding | 0.672 | 0.567 | 0.105 | 0.662 | 0.095 |
| | Death | 0.703 | 0.743 | -0.040 | 0.744 | 0.001 |
| High[c] | CV[d] | 0.831 | 0.802 | 0.029 | 0.835 | 0.033 |
| | Major bleeding | 0.706 | 0.635 | 0.071 | 0.703 | 0.068 |
| | Death | 0.852 | 0.849 | 0.003 | 0.848 | -0.001 |

[a] The difference between AUROC in the training vs validation set.

[b] Discrepancy between results based on EHR only vs. that based on the EHR-claims data.

[c] Low vs. high EHR-continuity cohorts are those with top 20% vs. lower 80% predicted EHR-continuity based on a published algorithm[3].

[d] Composite cardiovascular events (myocardial infarction or stroke).

AUROC = Area Under ROC Curve, NP = no preprocessing of data, LR = logistic regression, GB = gradient boosting, RF = random forest.

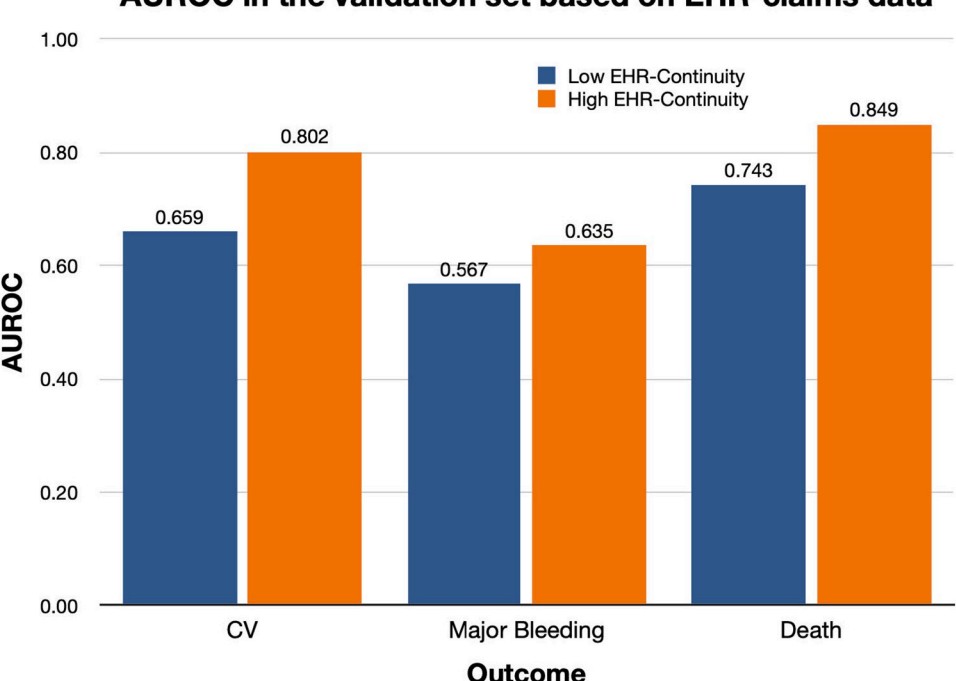

**Fig 1. AUROC in the validation set based on EHR-claims data.**

## Sensitivity analysis

We found a similar trend when using the AUPRC as the model performance metric. When predicting the same outcome, the AUPRC was greater in the high vs. low EHR continuity cohorts, which was consistent in both training and validation sets. For example, when predicting one-year major cardiovascular risk, the AUPRC was 0.523 vs. 0.352 in the high vs. low EHR continuity cohort in the training set and 0.481 vs. 0.372 in the validation set (*Table 4*).

## Discussion

Based on EHR from two large US metropolitan care delivery systems, we found that the prediction models developed among high EHR-continuity patients had consistently better

**Table 4. Area under the precision-recall curve and optimism by EHR-continuity.**

| EHR-continuity | Outcomes | AUPRC in the training set[a] | AUPRC in the validation set[a] |
|---|---|---|---|
| Low[b] | CV[c] | 0.352 | 0.372 |
| | Major bleeding | 0.169 | 0.189 |
| | Death | 0.166 | 0.205 |
| High[b] | CV[c] | 0.523 | 0.481 |
| | Major bleeding | 0.214 | 0.206 |
| | Death | 0.221 | 0.195 |

[a] Outcome events were assessed using EHR only in both training and validation set as prevalence of outcome is influential on AUPRC.

[b] Low vs. high EHR-continuity cohorts are those with top 20% vs. lower 80% predicted EHR-continuity based on a published algorithm[3].

[c] Composite cardiovascular events (myocardial infarction or stroke).

AUPRC = Area under the precision-recall curve, NP = no preprocessing of data, LR = logistic regression, GB = gradient boosting, RF = random forest.

performance and less optimism than models based on the low EHR-continuity cohort. The superior performance comparing the high vs. low EHR-continuity cohort was observed when predicting CV events, major bleeding, and mortality.

Our experiment set up the comparison to be between those with high vs. low EHR-continuity. The observed AUROC difference is thus mainly caused by difference in EHR-continuity. Because the input features were based on the same code-based definitions (see **S1 Table in S1 File**), the difference in model performance is likely due to different levels of misclassification of the input features. For instance, because the feature "Atrial Fibrillation (AF)" is based on having ICD-9 or ICD-10 diagnosis code of AF, this feature is more likely to be misclassified as "not present" for a patient with low EHR-continuity due to AF code recorded outside of the study EHR. We have previously observed up to 17-fold greater misclassification of commonly used CER variables, comparing those with the lowest (<10%) vs. highest (≥80%) level of EHR continuity [2]. When a prediction model is trained in those with low EHR-continuity, the predictors and outcomes may be substantially misclassified. Therefore, the associations identified by the models in the training set are not transportable to the validation set, potentially leading to reduced performance or greater optimism. We applied various ML methods and found a consistent magnitude of overfitting in the models developed among those with low EHR-continuity, suggesting these ML methods are equally susceptible to information bias due to EHR-discontinuity and such bias is unlikely to be mitigated in the modeling building process.

To overcome such bias, one solution is to link EHR data with additional data sources, such as insurance claims data. However, such linkage is often not feasible because of privacy (i.e., concerns for identification of the patients or breach of confidential information) and compliance issues (i.e., many data use agreements prohibit such linkage). Alternatively, investigators can use a previously published algorithm to identify those with high predicted EHR-continuity in the absence of linkage to additional data source [3]. Our analyses suggested that the performance of the models developed based on the top 20% of predicted EHR-continuity was significantly better than that developed based on those with lower EHR-continuity for 3 widely investigated clinical outcomes, mortality, cardiovascular and bleeding events.

One concern with restricting the analysis to patients in the high EHR-continuity cohort is that these patients may have a higher comorbidity burden and thus more frequent interactions with the healthcare system. However, prior studies have demonstrated there are no significant differences in comorbidity profiles among patients in high vs. low EHR-continuity levels when the profiles are measured using claims data that not susceptible to data-discontinuity [3]. This may be due to the fact that the patients who appear to have low EHR-continuity in a given EHR system may have high EHR-continuity in another EHR system (e.g., perhaps this patient only comes to the study EHR for specialist care and has the primary care physician in another EHR system). Therefore, having low EHR-continuity in one EHR system does not mean having infrequent interactions with the entire healthcare system or having a lower comorbidity burden in general. Nonetheless, it is acknowledged that there is a possible trade-off between generalizability (i.e., reduced ability to generalize findings derived in the high EHR-continuity cohort to the low EHR-continuity cohort) and internal validity (i.e., enhanced prediction accuracy and reduced optimism when the models were derived based on those with high EHR-continuity). Because nobody would want to "generalize" or apply biased results to other populations, we argue that the internal validity should take priority and building prediction models in the high EHR-continuity cohort is a reasonable approach.

There are several limitations in this study. First, our study population consists of adults aged 65 years or older and the findings may not be generalizable to the younger populations. Second, our results were based upon two urban EHR networks, which may be different from an EHR system of different demographics (e.g., systems serving a rural or less populated area).

Therefore, validation using different EHR systems is warranted. Third, limiting the study cohort to those with high EHR-continuity will inevitably reduce sample size, which may preclude building a rich and robust prediction model when the outcome events are infrequent.

In conclusion, we found that when predicting mortality, major CV and bleeding adverse outcomes in patients with CV comorbidities, the models developed using EHR data from patients with low EHR-continuity consistently had worse performance compared to that based on those with high EHR-continuity. Restriction of the prediction model development to those with high EHR-continuity could potentially enhance robustness of the findings.

## Supporting information

**S1 File. Supplemental Materials containing: Inclusion/Exclusion, covariate, and outcome variable definitions, EHR-continuity model, definitions of predictors of continuity, additional patient characteristic tables in the training and validation cohorts split by continuity status, additional method details, and follow-up time distribution.**
(DOCX)

## Acknowledgments

The authors would like to thank Ana Lyons, Ishan Khare, Lucy Cai, Anne Liang, Jamie Wells, and Victoria Wong for their comments on earlier drafts and Luke E. Zabotka for manuscript preparation.

## Author Contributions

**Conceptualization:** Shreyas Kar, Aaron S. Kesselheim, Kueiyu Joshua Lin.

**Data curation:** Shreyas Kar, Lily G. Bessette.

**Formal analysis:** Shreyas Kar, Lily G. Bessette, Richard Wyss, Kueiyu Joshua Lin.

**Funding acquisition:** Aaron S. Kesselheim, Kueiyu Joshua Lin.

**Investigation:** Shreyas Kar, Lily G. Bessette, Kueiyu Joshua Lin.

**Methodology:** Shreyas Kar, Lily G. Bessette, Richard Wyss, Kueiyu Joshua Lin.

**Project administration:** Kueiyu Joshua Lin.

**Software:** Shreyas Kar, Lily G. Bessette.

**Supervision:** Aaron S. Kesselheim, Kueiyu Joshua Lin.

**Validation:** Shreyas Kar, Lily G. Bessette, Richard Wyss.

**Visualization:** Shreyas Kar, Lily G. Bessette.

**Writing – original draft:** Shreyas Kar, Lily G. Bessette, Kueiyu Joshua Lin.

**Writing – review & editing:** Shreyas Kar, Lily G. Bessette, Richard Wyss, Aaron S. Kesselheim, Kueiyu Joshua Lin.

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
