## [Decision Letter · Decision Letter 0]

27 Dec 2022

PONE-D-22-25188The Impact of Electronic Health Record Discontinuity on Prediction ModelingPLOS ONE

Dear Dr. Lin,

Thank you for submitting your manuscript to PLOS ONE. After careful consideration, we feel that it has merit but does not fully meet PLOS ONE’s publication criteria as it currently stands. Therefore, we invite you to submit a revised version of the manuscript that addresses the points raised during the review process.

ACADEMIC EDITOR: Please revise and resubmit the manuscript. . 

We look forward to receiving your revised manuscript.

Kind regards,

Kathiravan Srinivasan

Academic Editor

PLOS ONE

Journal Requirements:

Reviewers' comments:

Reviewer's Responses to Questions

**Comments to the Author**

1. Is the manuscript technically sound, and do the data support the conclusions?

Reviewer #1: Yes

Reviewer #2: Yes

Reviewer #3: Partly

Reviewer #4: Yes

2. Has the statistical analysis been performed appropriately and rigorously? 

Reviewer #1: Yes

Reviewer #2: Yes

Reviewer #3: Yes

Reviewer #4: Yes

3. Have the authors made all data underlying the findings in their manuscript fully available?

Reviewer #1: No

Reviewer #2: No

Reviewer #3: Yes

Reviewer #4: No

4. Is the manuscript presented in an intelligible fashion and written in standard English?

Reviewer #1: Yes

Reviewer #2: Yes

Reviewer #3: Yes

Reviewer #4: Yes

5. Review Comments to the Author

Reviewer #1: It is not clear from the manuscript if you used EHR data (from Medicare) that was coded or did you somehow extract it from free text. Also, please explain the data in some more detail, including the fields used for the ML algorithms.

Reviewer #2: This paper is well organized and clearly written. The methodology and analysis are comprehensive, and the final conclusion is clearly summarized. The following issues should be addressed:

1. In the methodology, the data processing needs to be clarified: i.e., how to deal with categorical data and numerical data? One-hot vectors or binned numbers? These should be described in the Method or written with Table 1 when presenting the data features.

2. Though AUC scores are given, some insights are missing. For example, what features are causing such AUC difference? The chosen ML models may provide different weights to the input features, it is possible to show such analysis in the Discussion.

3. It may be possible to add bar charts instead of tables for better results presenting.

Reviewer #3: This paper applied machine learning algorithms based on EHR data in order to prove that when predicting outcomes like mortality, CV/bleeding events etc., patients with low EHR continuity consistently had worse performance than those who have high EHR continuity. The statement is demonstrated with logical supportive details. There are some problems need to be addressed to offer more clear presentation to readers.

1) From table 2, it seems that either lower or higher 50% of the events only have around 2000 event cases, compared to the cohort like like 32K in training, 20K in validation, which means your dataset is extremely imbalanced. It is better to show in details how did you use SMOTE to adjust that imbalance. Also, in the evaluation part, ROC AUC have weakness in showing true performance of imbalanced classification task, it's strongly recommended to add precision recall AUC and show AUC difference between high/low EHR continuity.

2)It's not clearly described that how EHR features are formatted in the predictive models. For example, whether dummy variables are generated for categorial predictors? In the medical history, in addition to indicators of diagnosed diseases, whether there are other information can be extracted? Like time interval between diagnosis and events? Also, whether there are treatments or drug use details can be used as predictors?

3)2007 to 2017 is a very wide time range, what time periods are the training and validation cohort data are obtained specifically? Also, it is better to show the patient duration distribution in training/validation like minimal/median/maximum time that a patient can be tracked in the dataset.

4)More introduction can be added to explain the definition and extent of EHR-continuity.

Reviewer #4: The authors examine the impact of EHR continuity on the performances of machine learning algorithms to predict three widely investigated clinical outcomes. The authors used an existing model to classify if a patient experienced EHR discontinuity, and the training set was split into two parts, which have either high EHR continuity or low EHR continuity. Then ML models are trained separately using these two parts to report and compare results on an external validation set. The experimental setup of the study is commendable and the statistical analysis is sound, presenting sufficient details to support its conclusion on the impact of EHR continuity on ML models. I have two major comments.

1. EHR continuity was predicted using an off-the-shelf algorithm. I think a bit more detail on the evaluation of this algorithm would better inform the readers. The algorithm can also be viewed as a filter, and we see improved downstream performances after removing the “noisy” half of the training data using the algorithm. This makes me wonder if any other factors might result in the difference. For example, is it possible that 50% of data with low EHR continuity also suffers from a higher level of data missingness? These two are obviously correlated but not equivalent since there could be other causes of data missingness other than EHR discontinuity. Maybe more descriptive statistics on the two datasets with varied EHR continuity would be helpful.

2. The authors developed models using two datasets with varied EHR continuity and evaluated them on the unified validation set. It may be worth considering also splitting the validation set into two parts according to EHR continuity, similar to the development set. Maybe some supplementary results here would better cast light on the impact of EHR continuity and whether the impact is consistent throughout training and testing. Intuitively, both models developed regardless of EHR continuity should perform better on the validation set with higher continuity than on that with lower continuity. But if the model developed using low EHR-continuity data also performs better on data with low rather than high continuity, then maybe distributional shift plays a more significant factor in the different modeling results. In other words, are the development and validation sets equally “continuous”? Would that matter?

One minor comment on data sharing: the authors should specify more explicitly the restrictions preventing them from sharing the data.

6. PLOS authors have the option to publish the peer review history of their article (what does this mean?). If published, this will include your full peer review and any attached files.

Reviewer #1: No

Reviewer #2: No

Reviewer #3: No

Reviewer #4: No

---

## [Author Response · Author response to Decision Letter 0]

26 May 2023

Response to reviewer comments are included in the Response to the Reviewers_PLOS ONE_FINAL.docx

---

## [Decision Letter · Decision Letter 1]

19 Jun 2023

The Impact of Electronic Health Record Discontinuity on Prediction Modeling

PONE-D-22-25188R1

Dear Dr. Lin,

We’re pleased to inform you that your manuscript has been judged scientifically suitable for publication and will be formally accepted for publication once it meets all outstanding technical requirements.

Kind regards,

Kathiravan Srinivasan

Academic Editor

PLOS ONE

Additional Editor Comments (optional):

Reviewers' comments:

Reviewer's Responses to Questions

**Comments to the Author**

1. If the authors have adequately addressed your comments raised in a previous round of review and you feel that this manuscript is now acceptable for publication, you may indicate that here to bypass the “Comments to the Author” section, enter your conflict of interest statement in the “Confidential to Editor” section, and submit your "Accept" recommendation.

Reviewer #1: All comments have been addressed

Reviewer #3: All comments have been addressed

Reviewer #4: All comments have been addressed

2. Is the manuscript technically sound, and do the data support the conclusions?

Reviewer #1: (No Response)

Reviewer #3: Yes

Reviewer #4: Yes

3. Has the statistical analysis been performed appropriately and rigorously? 

Reviewer #1: (No Response)

Reviewer #3: Yes

Reviewer #4: Yes

4. Have the authors made all data underlying the findings in their manuscript fully available?

Reviewer #1: (No Response)

Reviewer #3: No

Reviewer #4: No

5. Is the manuscript presented in an intelligible fashion and written in standard English?

Reviewer #1: (No Response)

Reviewer #3: Yes

Reviewer #4: Yes

6. Review Comments to the Author

Reviewer #1: (No Response)

Reviewer #3: The authors have address all my previous comments. And it reads clearly to me now. The table layout and fonts are not consistent, hopefully it can be well addressed before publishing.

Reviewer #4: Thank you for addressing the raised the comments and updated the manuscript accordingly, which has been substantially improved.

7. PLOS authors have the option to publish the peer review history of their article (what does this mean?). If published, this will include your full peer review and any attached files.

Reviewer #1: No

Reviewer #3: No

Reviewer #4: No

---

## [Editor Report · Acceptance letter]

22 Jun 2023

PONE-D-22-25188R1 

The Impact of Electronic Health Record Discontinuity on Prediction Modeling 

Dear Dr. Lin:

I'm pleased to inform you that your manuscript has been deemed suitable for publication in PLOS ONE. Congratulations! Your manuscript is now with our production department. 

Kind regards, 

on behalf of

Dr. Kathiravan Srinivasan 

Academic Editor

PLOS ONE